# TCF Plus Radiochemotherapy Versus Neoadjuvant Radiochemotherapy Versus Flot Perioperative Chemotherapy in Esophageal Adenocarcinoma: The Results of a Three-Cohort, Multi-Centric Comparison: The A4 Study

**DOI:** 10.3390/biomedicines13092236

**Published:** 2025-09-11

**Authors:** Marco Lorenzo Bonù, Giulia Volpi, Gloria Zanni, Jacopo Balduzzi, Fabrizia Terraneo, Giusto Pignata, Giuseppina Arcangeli, Francesco Frassine, Paola Vitali, Eliana La Rocca, Simone Giacopuzzi, Jacopo Weindelmayer, Carlo Alberto De Pasqual, Martina Milazzo, Michele Pavarana, Valentina Zen, Stefano De Pascale, Uberto Fumagalli Romario, Michela Buglione, Giovanni De Manzoni

**Affiliations:** 1Department of Radiation Oncology, Asst Spedali Civili di Brescia, 25100 Brescia, Italy; marco.bonu@unibs.it (M.L.B.); g.zanni003@studenti.unibs.it (G.Z.); j.balduzzi@unibs.it (J.B.); fabriziaterraneo@virgilio.it (F.T.); francesco.frassine@asst-spedalicivili.it (F.F.); paola.vitali@asst-spedalicivili.it (P.V.); buglione@med.unibs.it (M.B.); 2Radiation Oncology, Azienda Ospedaliera Universitaria Integrata di Verona, 37126 Verona, Italy; eliana.larocca@aovr.veneto.it; 3Department of Surgery, Asst Spedali Civili di Brescia, 25100 Brescia, Italy; giusto.pignata@asst-spedalicivili.it; 4Department of Medical Oncology, Asst Spedali Civili di Brescia, 25100 Brescia, Italy; giuseppina.arcangeli@asst-spedalicivili.it; 5General and Upper GI Surgery Division, University of Verona, Borgo Trento, 37124 Verona, Italy; simone.giacopuzzi@univr.it (S.G.); jacopo.weindelmayer@aovr.veneto.it (J.W.); carloalberto.depasqual@aovr.veneto.it (C.A.D.P.); martina.milazzo@studenti.univr.it (M.M.); giovanni.demanzoni@univr.it (G.D.M.); 6Department of Oncology, University of Verona Hospital Trust, 37126 Verona, Italy; michele.pavarana@aovr.veneto.it (M.P.); valentina.zen@studenti.univr.it (V.Z.); 7Digestive Surgery, European Institute of Oncology IRCCS, 20141 Milan, Italy; stefano.depascale@ieo.it (S.D.P.); umberto.fumagalliromario@ieo.it (U.F.R.)

**Keywords:** esophageal adenocarcinoma, radiochemotherapy, neoadjuvant therapies, perioperative chemotherapy

## Abstract

**Introduction:** Recent randomized evidence suggests that stage II–IV non metastatic esophageal adenocarcinoma is best managed with perioperative chemotherapy (CHT) and surgery. Intensification of neoadjuvant chemotherapy and radiochemotherapy are proposed before surgery in high-volume centers with the aim of increasing both systemic and locoregional control. However, few data comparing intensified RTCHT, CHT plus RTCHT and perioperative CHT with FLOT in real-life scenarios are available. **Methods:** This is a multicenter, retrospective series, including three cohorts of patients treated for esophageal adenocarcinoma: Cohort A: nRTCHT; Cohort B: TCF plus RTCHT, defined as triplet chemotherapy followed by dose-reduced triplet therapy + RT; Cohort C: perioperative chemotherapy with FLOT regimen. The primary endpoint was disease-free survival (DFS), and the secondary endpoints were pathologic complete response (pCR), pathologic lymph-node complete response (ypN0), overall survival (OS), and perioperative acute toxicity. **Results:** From January 2013 to December 2023, 142 patients were identified. All patients received multimodal therapy with radical esophagectomy. A total of 95% of patients were male; the majority of patients presented with stage cT3cN1. A total of 63 patients were treated in Cohort A (31 cases with doublet 5FU-CDDP concurrent to 50.4 Gy and 32 cases with CROSS regimen), 36 in Cohort B, and 43 in Cohort C. After a median FU of 36 months, the 3-year DFS resulted 58.6%. pCR occurred in 26 cases (18.6%). Three-year OS had a value of 72%. At univariate analysis, ypN0 was related to better DFS; cN+ disease was related with worse OS. The treatment cohort did not impact survival outcomes; however, an effect on CR was shown, with pCR in 15% (A), 36.3% (B), 11% (C) of cases, respectively (χ: 0.008). A total of 67% of patients in Cohort B experienced a ypN0. Two treatment-related deaths occurred (one in Cohort A and one in C) with a slight increase in G3 toxicity in cohort C. **Conclusions:** In this real-life multicenter series, oncological results were adequate for all three neoadjuvant strategies. TCF plus RTCHT guaranteed a higher pCR and ypN0 rate without increasing toxicity. An intensified neoadjuvant schedule, such as TCF plus RTCHT, may be useful in cases where higher tumor and nodal responses are needed. Taken together, our data highlight that further investigation is warranted before abandoning radiotherapy-based neoadjuvant approaches in esophageal and GEJ adenocarcinoma.

## 1. Introduction

Neoadjuvant radiochemotherapy (nRTCHT) followed by surgical resection has been the standard of care for resectable stage II–IV non-metastatic esophageal adenocarcinoma for over a decade. This approach was primarily established based on the findings of the CROSS trial and its subsequent subgroup analyses [1].

However, evidence revealed certain limitations of the CROSS regimen in effectively controlling distant metastatic spread in real-life scenarios [2].

In parallel, the perioperative chemotherapy regimen FLOT (5-FU, leucovorin, oxaliplatin, and docetaxel), which has demonstrated superior outcomes in gastric cancer, has emerged as a promising alternative for esophageal adenocarcinoma. This prompted investigations into the comparative efficacy of FLOT versus CROSS in the esophageal cancer setting. The Neo-AEGIS trial suggested non-inferiority between the two regimens, while the more recent ESOPEC trial reported a significant advantage of FLOT over CROSS across multiple oncological endpoints, including overall survival and pathological response, but not for complete nodal response, which remained slightly better in nRTCHT. Notably, the CROSS arm in this trial underperformed compared to expectations and previous observations [3,4].

To further improve both locoregional and systemic disease control, preoperative triplet chemotherapy followed by radiochemotherapy and intensified neoadjuvant radiochemotherapy (iRTCHT) are being increasingly explored in high-volume centers and in subgroups of relevant clinical trials for gastric cancers, also enrolling those with gastro-esophageal junction (GEJ) Siewert I and II disease [5,6,7].

These strategies aim to overcome the limitations of existing regimens by optimizing tumor response prior to surgery. Despite their growing adoption, there is limited real-world data directly comparing intensification of neoadjuvant chemotherapy plus nRTCHT, nRTCHT alone, and perioperative FLOT in patients with localized esophageal adenocarcinoma.

To address this gap, we conducted a cohort study, evaluating three contemporary approaches—neoadjuvant chemotherapy followed by RTCHT, standard nCRT, and perioperative FLOT—in a real-world clinical setting.

## 2. Materials and Methods

### 2.1. Study Design and Patient Selection

This multicenter retrospective cohort study was conducted at two reference centers located in Northern Italy. The study included patients meeting the following criteria:Histologically confirmed esophageal adenocarcinoma (clinical stage cT2–T4a or any T with cN+), including gastroesophageal junction (GEJ) tumors classified as Siewert types I–II.Pre-treatment staging with contrast-enhanced computed tomography (CT), esophagogastroduodenoscopy, and positron emission tomography–computed tomography (PET-CT).Treatment received in one of the following three cohorts:○Cohort A: nRTCHT as per CROSS regimen (41.4 Gy in 23 fractions with concurrent chemotherapy with intravenous area-under-the-curve 2 mg/mL per min carboplatin plus intravenous 50 mg/m^2^ paclitaxel on days 1, 8, 15, 22, and 29) or an intensified RTCHT regimen consisting of cisplatin 75 mg/m^2^ on days 1 and 29 and 5-fluorouracil (5-FU) 4000 mg/m^2^ as a continuous infusion on days 1–4 and 29–33.○Cohort B: Triplet chemotherapy followed by nRTCHT, defined as triplet chemotherapy using the weekly TCF regimen (docetaxel 35 mg/m^2^ and CDDP 25 mg/m^2^ on days 1, 8, and 15, plus 5FU 180 mg/m^2^/day continuous infusion on days 1 to 21) followed by dose-reduced weekly TCF combined with radiotherapy (for CHT: docetaxel 35 mg/m^2^ and CDDP 25 mg/m^2^ weekly during RT and 5FU 150 mg/m^2^/day continuous infusion on days 29 to 63; for RT: 41.4 Gy at 1.8 Gy per fraction, with a sequential boost to macroscopic FDG-PET–avid residual disease after induction chemotherapy up to a total dose of 50.4 Gy), the so-called “Verona Regimen”.○Cohort C: Perioperative chemotherapy based on the FLOT regimen consisting in four preoperative and four postoperative 2-week cycles of 50 mg/m^2^ docetaxel, 85 mg/m^2^ oxaliplatin, 200 mg/m^2^ leucovorin and 2600 mg/m^2^ fluorouracil as 24 h infusion on day 1.

After neoadjuvant therapy, patients were re-staged using CT, esophagogastroduodenoscopy, endoscopic ultrasound, and PET-CT, with the specific modalities used at the discretion of the treating physician. All patients subsequently underwent radical esophagectomy using the Ivor Lewis approach, performed via either laparoscopic/thoracoscopic, laparotomic/thoracotomic, or robotic techniques.

Exclusion criteria were age under 18 years, Eastern Cooperative Oncology Group performance status (ECOG PS) ≥ 3, initially unresectable disease, and presence of metastatic disease at diagnosis.

The study adhered to the principles of the Declaration of Helsinki and was approved by the Ethical Committee of the Coordinating Center. All data were anonymized and collected using an electronic case report form (CRF). Recorded variables included patient demographics, clinical status, weight loss before and after induction treatment, TNM classification, tumor location, chemotherapy regimen, radiotherapy dose and technique, treatment timelines, toxicity (graded according to CTCAE v5.0), surgical details, pathological response, disease control, recurrence patterns, overall survival (OS), and cause of death. The database was validated to ensure data accuracy, and mandatory fields were implemented to minimize missing information. All statistical analyses were conducted on anonymized datasets.

### 2.2. Definition of Study Endpoints

Disease-Free Survival (DFS): Time from surgery to recurrence at any site or last follow-up.Distant Metastasis-Free Survival (DMFS): Time from surgery to the occurrence of distant metastasis or last follow-up.Freedom from Local Recurrence (FFLR): Time from surgery to local recurrence at the site of prior RTCHT or surgery, or last follow-up.OS: Time from surgery to death from any cause or last follow-up.Postoperative Acute Toxicity: Defined as any non-hematologic adverse event occurring between the date of surgery and the 90th postoperative day, graded according to CTCAE v5.0.

### 2.3. Follow-Up and Statistical Analysis

Postoperative follow-up consisted of total-body CT and esophagogastroduodenoscopy performed every six months or the first three years, then annually up to the seventh year post-treatment.

Descriptive statistics were used to summarize the collected data. Estimates were presented with corresponding 95% confidence intervals. Inferential analyses were performed using the Chi-square (χ^2^) test to assess differences in tumor stage, nodal status, age, ECOG PS, pathologic complete response (pCR), pathologic nodal complete response (ypN0), and treatment compliance across the three cohorts.

Univariate analysis (UVA) for survival endpoints was conducted using the Kaplan–Meier method, with the log-rank test employed to compare survival distributions. Evaluated covariates included ECOG PS, age, tumor site, tumor grade (G1–2 vs. G3), clinical T and N stage, overall stage group, treatment cohort, compliance with induction therapy (defined as administration of at least 70% of the planned chemotherapy dose and at least 95% of the planned radiotherapy dose without interruptions exceeding 5 days), radiotherapy technique, surgical approach, tumor response after neoadjuvant therapy, pCR, ypN0, and preoperative weight loss (>10% vs. ≤10% of baseline body weight). Statistical analyses were performed using IBM SPSS Statistics software, version 30.0 (IBM Corp., Armonk, NY, USA). A *p*-value < 0.05 was considered statistically significant. Multivariate analysis was not conducted due to the results observed in the UVA.

## 3. Results

Between January 2013 and December 2023, a total of 142 patients were included in the study. The majority of tumors were located in the distal esophagus (n = 55, 39%), followed by Siewert type I (n = 37, 26%) and Siewert type II tumors (n = 50, 35%). All patients received multimodal treatment including radical esophagectomy. The study population was predominantly male (95%).

Most patients presented with clinical stage cT3cN1 at diagnosis. Of the total population, 63 patients were treated in Cohort A (n = 31 received the intensified 5-FU/CDDP regimen with concurrent 50.4 Gy RT, and n = 32 received the CROSS protocol), 36 in Cohort B, and 43 in Cohort C. Detailed baseline patients characteristics are summarized in Table 1.

Patients in Cohort A more frequently exhibited an ECOG PS of 2, while those in Cohorts B and C had a higher prevalence of cN1 disease. Compliance to CHT and RT was well balanced between the cohorts, with approximately 95% of patients that completed the prescribed RT schedule in Cohort A and 92% in Cohort B. Of the 77 patients (Cohort B plus C) treated with induction/perioperative chemotherapy, compliance was less than 70% in 35% of cases.

### 3.1. Pathological Response

A pathological complete response (pCR) was observed in 27 patients (18.5%) across the study population. The distribution of pCR by cohort was as follows:Cohort A: 15%;Cohort B: 36.1%;Cohort C: 11%.

The difference in pCR rates among the cohorts was statistically significant in favor of Cohort B (χ^2^ test, *p* = 0.008).

Pathologic nodal complete response (ypN0) was achieved in 67% of patients in Cohort B, compared to 45.8% in Cohort A and 53.8% in Cohort C; (*p* = not significant). The distribution of relevant clinicopathological variables across cohorts is presented in Table 2.

### 3.2. Survival Outcomes

At a median follow-up of 36 months for alive patients (32 months for the whole series), the estimated 3-year DFS for the overall cohorts was 58.6%, with a median DFS of 43 months. The 3-year OS, DMFS, and FFLR rates were 72%, 61.3%, and 83%, respectively.

UVA results for survival endpoints are summarized below. Loss of follow up were acceptable; concerning OS, data were available for 98% of patients. For alive patients, concerning DFS, FFLR, FFDM 95% of patients completed regularly 12 months of follow-up, 92% of patients completed regularly 36 months of follow-up.

For the primary endpoint of the study, three-year DFS showed percentages of 54.4%, 53.3%, 63% in Cohort A, B and C, respectively (*p* = ns). The achievement of a ypN0 was significantly associated with improved DFS (*p* = 0.004). Additionally, a trend toward improved DFS was observed in patients with pre-treatment weight loss ≤10% of baseline body weight (*p* = 0.053).

Figure 1a,b illustrates Kaplan–Meier survival curves with corresponding log-rank tests for DFS across treatment cohorts and stratified by ypN0 status.

Three-year OS showed values of 72%, 76%, and 68% for Cohort A, B, and C, respectively (*p* = ns). Univariate analysis of OS demonstrated a significantly better outcome in patients with clinically node-negative (cN0) disease (*p* < 0.001). A trend toward improved OS was also observed in patients achieving pathologic nodal complete response (ypN0), although this did not reach statistical significance (*p* = 0.071). Detailed univariate analysis results concerning three-year DFS and OS are presented in Table 3.

Regarding FFLR, improved outcomes were associated with ECOG performance status of 0 or 1 (*p* = 0.004), pre-treatment weight loss ≤ 10% of baseline body weight (*p* = 0.001), and the presence of ypN0 (*p* = 0.001).

No variables were found to significantly impact distant metastasis-free survival (DMFS); however, a trend toward better DMFS was observed in patients without significant weight loss and those with ypN0.

Subgroup analysis of Cohort A concerning patients treated with CROSS scheme (31 patients) and patients treated with 50.4 Gy + 5FU and CDDP (32 patients) did not showed any significant difference on each survival outcome.

### 3.3. Treatment-Related Toxicity

Two cases of treatment-related mortality (grade 5 toxicity) occurred within 90 days post-esophagectomy. One was reported in Cohort A and one in Cohort C.

In Cohort A, one case involved a 75-year-old male treated with the CROSS regimen who developed intestinal ischemia and subsequent multi-organ failure on postoperative day 3. In Cohort C, a 77-year-old male developed an anastomotic leak complicated by pneumonia and respiratory failure, ultimately resulting in death on the 20th postoperative day in spite of intensive care and mechanical ventilation.

Non-fatal grade 4 toxicities were observed in four patients in Cohort A, one in Cohort B, and three in Cohort C. These events were predominantly related to severe septic, cardiac, or pulmonary complications.

Grade 3 toxicities were documented in 12, 6, and 11 patients in Cohorts A, B, and C, respectively. The most common complications included anastomotic leakage requiring invasive management, new-onset atrial fibrillation, and pulmonary complications such as pneumonia or pleural effusion. Of the 12 non-fatal leakages that occurred in our series, 3 were managed with re-operation, 5 with endoscopic therapy such as stenting, and 4 with transluminal vacuum therapy.

A comprehensive summary of acute postoperative toxicity profiles across all three cohorts is provided in Table 4. Moreover, a report of hematological toxicity for the three cohorts is provided in Appendix A, while preoperative and postoperative detailed G ≥ 2 toxicity regarding cohorts is provided in Appendix A.

## 4. Discussion

The present investigation expands the debate about the optimally integrating systemic and locoregional treatment for resectable esophageal and Siewert I–II junctional adenocarcinoma. All three neoadjuvant approaches have shown comparable long-term outcomes both in terms of DFS and OS, comparable or better than the one reported in contemporary randomized trials, such as the 57% three-year OS of both arms in Neo-AEGIS and the 57% three-year OS of the FLOT arm in ESOPEC, achieving more favorable results in Cohort A in terms of three-year OS [3,4].

Although no evidence of superiority in outcomes was shown in UVA by treatment cohort, the regimens displayed significatively different biological activity. TCF plus RTCHT achieved a pCR of 36% and ypN0 in two-thirds of patients, doubling the rates achieved by peri-operative FLOT and standard CROSS in our hands. pCR is generally associated with excellent prognosis in esophageal cancer [8,9] and in the literature, ypN0 is associated with better OS and DFS compared to those with residual nodal disease [10,11]. Our UVA confirmed that patients achieving nodal complete response had significantly improved DFS. Importantly, achieving pCR presents relevant implications, facilitating surgery, especially in bulk cT3 and complex cT4 cases, and indicating robust tumor radiochemosensitivity [12,13].

The TCF plus RTCHT regimen appears to have maximized the cytotoxic impact on the primary tumor and regional nodes—yielding a pCR rate (36%) that is among the highest reported for EAC in a multimodal therapy context. By comparison, the CROSS trial reported a pCR of ~23% in adenocarcinomas and perioperative chemotherapy alone typically yields pCR between 7 and 9% in non-German studies [14]. Despite this enhanced tumor regression, the improvement did not translate into superior DFS or OS for the TCF plus RTCHT group. These findings mirror the recent results of the phase III Neo-AEGIS trial, that reported similar 3-year OS (55% vs. 57%) between the chemotherapy and chemoradiation arms, even though nRTCHT led to higher rates of pCR, major pathologic response, R0 resection, and nodal downstaging. The reasons underlying our favorable results across the three cohorts are multifaceted. They may, in part, relate to the per-protocol use of baseline PET staging, which could have facilitated the detection of metastatic disease occult on CT scans and thereby allowed for a more accurate selection of patients with true locoregional disease. This represents a clear limitation of the ESOPEC trial, in which PET-CT was not employed for staging purposes. Furthermore, our series excluded non-responders to induction therapy as well as patients who did not proceed to esophagectomy (e.g., due to disease progression or deterioration of clinical condition). This approach may have contributed to the selection of a patient population with a more favorable prognosis.

Peri-operative FLOT, validated by the FLOT4-AIO trial [15] and by later real-world analyses [16,17], is attractive because it delivers the highest cumulative dose of systemic chemotherapy. ESOPEC showed a particularly strong survival advantage of FLOT over CROSS (median OS 66 vs. 37 months), yet several caveats limit the generalizability of the findings. In addition to the aforementioned limitation regarding the lack of per-protocol PET staging—which may have concealed a potential imbalance in occult metastatic spread between trial arms—it should be noted that a slight majority of patients in the CROSS arm were cN+, and 11 patients did not start RTCHT due to the occurrence of metastatic disease. Moreover, the pCR rate was unusually low (10%), raising some doubt as to RT quality assurance. Last but not least, no postoperative immunotherapy was permitted. Such considerations contribute to underlining the need to exercise caution in interpreting the trial results. Conversely, national series have shown that when CROSS is delivered with strict adherence, real-world pCR and three-year OS are comparable to those obtained with CROSS original arm and peri-operative chemotherapy [2].

Remarkably, Cohort C failed to outperform the other strategies in our study. The reasons for this are multifaced. First, the fair results of other cohorts clearly reduce the potential gap of efficacy evident in ESOPEC, but other reasons may lie in the complexity of delivery “per protocol” FLOT schedule in real-life scenarios, where patient compliance and treatment side effects are an issue in maintaining dose intensity. Grade ≥ 3 neutropenia and gastrointestinal events were slightly more frequent in the FLOT cohort, consistent with observational reports [18]. Furthermore, postoperative cycles are administered after surgical stress, whereas TCF plus RTCHT front-loads systemic therapy, potentially enhancing compliance, a fact that is also evident in ESOPEC were only 54% of patients to receive the four adjuvant cycles.

Toxicity considerations are crucial to real-world feasibility. Notwithstanding its intensity, TCF plus RTCHT did not increase peri-operative mortality and showed a composite toxicity profile overlapping the other two regimens.

Regarding acute adverse events, patients in the perioperative chemotherapy cohort experienced slightly more grade 3 toxicities in our series, consistent with the known profile of FLOT (e.g., neutropenia, diarrhea, neuropathy). Published real-world data have similarly shown that FLOT-based therapy is less well tolerated, with higher rates of ≥grade 3 events and the need for dose modifications, compared to the CROSS chemoradiation regimen [18].

nRTCHT remains a reasonable option when a short, well-tolerated approach is needed—such as in patients unfit for triplet chemotherapy, those with a potential risk of positive resection margins, or in contexts where postoperative therapy is unlikely to be completed—especially now that residual disease can be targeted with one year of nivolumab [19]. Peri-operative FLOT may best suit younger, robust patients in centers experienced in dose-intense chemotherapy, whereas triplet chemotherapy plus RTCHT seems an attractive compromise for biologically aggressive, bulk, borderline-resectable, node-positive tumors or for patients unlikely to tolerate postoperative chemotherapy. By front-loading therapy, it is possible to maximize compliance, to achieve the deepest pathological responses, and to allow functional recovery to begin immediately after resection. These remain hypotheses that warrant prospective confirmation rather than prescriptive rules.

Our study presents some limitations. Treatment allocation was not randomized, with an evident tendency to offer TCF plus RTCHT and FLOT to patients with better ECOG PS; on the other hand, Cohort B and C show also the tendency to have more advanced disease concerning both T and N stage. Some heterogeneity exists also within the CRT cohort (Cohort A), as it included both standard CROSS and an “intensified” cisplatin/5-FU CRT regimen; although our subgroup analysis did not find outcome differences between these CRT subgroups, this intra-cohort variation could obscure subtle effects. Moreover, as a multi-institutional retrospective study, there may have been variability in supportive care, surgical techniques, and pathologic assessment.

This study’s strengths include the accrual of patients at two reference centers over a decade, reflecting contemporary practice in EAC, including the intensified triplet CHT plus RTCHT regimen rarely described in the literature. Given that most recurrences in esophageal adenocarcinoma occur within 2–3 years post-resection, a median follow-up of 36 months should be sufficient to assess relevant survival endpoints.

Similarly to our study context, there is a RACE trial comparing perioperative FLOT vs. FLOT + neoadjuvant RTCHT, and it will possibly elucidate whether an intensified regimen of both chemo and radiotherapy may result in a benefit on outcomes within a randomized context [20]. Moreover, perioperative chemo-immunotherapy trials [21,22] along with the eagerly awaited OS results of CheckMate 577 trial may further analyze the impact of checkpoint inhibitors in the optimization of systemic disease control. Finally, refining patient selection through molecular and clinical predictors of response will potentially direct each patient to the regimen most likely to benefit, narrowing the gap between chemo-only and chemo-radiation strategies [23,24].

## 5. Conclusions

Our multi-center, real-life experience shows that neoadjuvant TCF plus RTCHT, neoadjuvant chemoradiation, and perioperative chemotherapy all achieve fair oncologic results in localized EAC. Nevertheless, our cohorts exhibit differences in pCR, ypN0, and a toxicity profile that seems to favor TCF plus RTCHT. Perioperative FLOT is also an effective strategy in our multicenter retrospective scenario; nevertheless, in large/bulky tumors in which downstaging becomes relevant before surgery, RTCHT could be a valid option, especially with more systemically active regimens. Taken together, our data highlights that further investigation is warranted before abandoning radiotherapy-based neoadjuvant approaches in esophageal and EGJ adenocarcinoma.

## Figures and Tables

**Figure 1 biomedicines-13-02236-f001:**
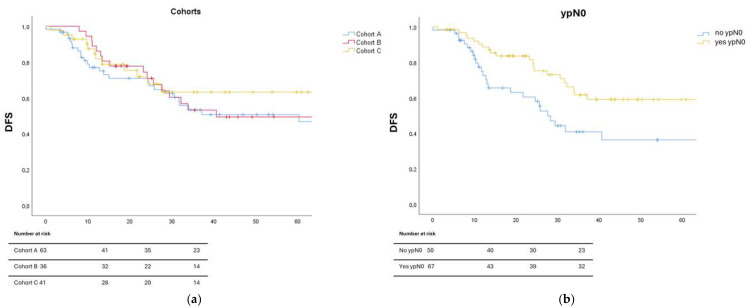
Kaplan–Meier curve representing the log-rank test for DFS in the three cohorts (**a**) and for pathologic lymph node complete response achieved or not on definitive histopathologic exam (**b**).

**Table 1 biomedicines-13-02236-t001:** Patient characteristics in the whole series.

Variable	Number (%)
ECOG PS	
0-1	123 (87)
2	19 (13)
Median AGE (range)	62 (range 28-81)
TUMOR SITE	
Distal esophagus	55 (39)
Siewert I	37 (26)
Siewert II	50 (35)
Tumor grade	
G1-2	79 (56)
G3	63 (44)
cT stage	
cT2	13 (9)
cT3	118 (83)
cT4	36 (25)
cN stage	
cN0	25 (18)
cN+	117 (82)
Stage group	
II	24 (17)
III	107 (75)
IVA	11 (8)
Compliance to systemic chemotherapy (tot 77)	
No	27 (35)
yes	50 (65)
Compliance to radiotherapy (tot 99)	
No	6 (6)
yes	93 (94)
Radiotherapy technique (tot 99)	
3D	7 (7)
IMRT	17 (17)
VMAT	30 (30)
HELICAL IMRT	45 (46)
Surgical approach	
Open	16 (11)
Minimally invasive abdominal	14 (10)
Minimally invasive thoracic and abdominal	112 (79)
Clinical response before surgery	
CR	4 (3)
PR	119 (84)
SD	13 (9)
Local or regional PD	3 (2)
pCR	
no	115 (82)
yes	27 (18)
ypN0	
no	76 (54)
yes	66 (46)
Weight loss pre-therapy	
≤10% of baseline	97 (68)
>10% of baseline	45 (32)

**Table 2 biomedicines-13-02236-t002:** Distribution of the variables of interest between the cohorts and Χ^2^ results.

Variable	Cohort A n° (%)	Cohort B n° (%)	Cohort C n° (%)	Χ^2^ (*p* Value)
ECOG PS				
0-1	52 (82%)	35 (97%)	36 (84%)	0.095
2	11 (18%)	1 (3%)	7 (16%)	
Median AGE (range)	62 (43–80)	61 (50–81)	66 (28–77)	0.6
cT stage				
cT2	9 (15%)	1 (3%)	3 (7%)	0.2
cT3	50 (79%)	34 (94%)	36 (84%)	
cT4	4 (6%)	1 (3%)	4 (9%)	
cN stage				
cN0	17 (27%)	4 (11%)	4 (9%)	0.033
cN+	46 (73%)	32 (89%)	39 (91%)	
pCR				
no	54 (85%)	23 (64%)	38 (89%)	0.008
yes	9 (15%)	10 (36%)	5 (11%)	
ypN0 (tot 117)				
no	26 (57%)	11 (33%)	17 43%)	0.1
Yes	20 (43%)	21 (67%)	22 (57%)	

**Table 3 biomedicines-13-02236-t003:** Detailed univariate analysis concerning three-year DFS and OS.

Variable	3 ys DFS (%)	*p* Value	3 ys OS (%)	*p* Value
Cohort		ns		ns
A	54.4		72.7	
B	53.3		76	
C	63		67.8	
ECOG PS		ns		ns
0-1	49.1		62.9	
2	56.1		73.4	
cT stage		ns		ns
cT2	65.5		83.9	
cT3	55.7		70.4	
cT4	41.5		76.2	
cN stage		ns		<0.1
cN0	69.2		82.7	
cN+	52		70.3	
pCR		ns		ns
no	53.2		72	
yes	62.4		73	
ypN0 (tot 117)		0.4		0.05
no	38.7		62.6	
yes	62.4		77.7	
Pre-treatment weight loss		0.053		ns
≤10% of baseline	59.6		78.9	
>10% of baseline	38.2		71.8	

Legenda: ns = not significant.

**Table 4 biomedicines-13-02236-t004:** Postoperative acute non-hematological toxicity.

Cohort	G0	G1	G2	G3	G4	G5	Tot
A (RTCHT)	18 (29%)	7 (11%)	20 (32%)	12 (19%)	4 (6%)	1 (1.5%)	63
B (TCF plus RTCHT)	19 (53%)	1 (3%)	9 (25%)	6 (16%)	1(3%)	0 (0%)	36
C (FLOT)	17 (41%)	2 (5%)	7 (17%)	11 (27%)	3(7%)	1 (2.4%)	41

## Data Availability

The original contributions presented in this study are included in the article/Appendix A. Further inquiries can be directed to the corresponding author.

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
