# Peer review of "TCF Plus Radiochemotherapy Versus Neoadjuvant Radiochemotherapy Versus Flot Perioperative Chemotherapy in Esophageal Adenocarcinoma: The Results of a Three-Cohort, Multi-Centric Comparison: The A4 Study"

_biomedicines, 2025, doi:10.3390/biomedicines13092236_

Round 1
Reviewer 1 Report
Comments and Suggestions for Authors
- Introduction: In the second paragraph, the author states, " However, these studies also highlighted certain limitations of the CROSS regimen." However, no specific references are cited to identify which studies reported these limitations of the CROSS regimen.
- Introduction: The fourth paragraph is missing a full stop.
- Introduction: The full name of GEJ should be written out when it first appears, rather than when it is mentioned for the second time in the Methods.
- Methods: The approval number and date from the ethics committee were not provided.
- Methods: Please confirm whether the description of radiotherapy as "RT: 41.4 Gy in 1.8 Gy fractions" is accurate.
- Results: Only 142 patients were included in two high-volume centers over a period of 10 years. The overall sample size is far too small to justify the claim that these are high-volume centers.
- Results: As a study that began in 2013, the median follow-up period was only 36 months. What were the longest and shortest overall follow-up periods? The follow-up methodology was described in the Methods section as follows: “Postoperative follow-up consisted of total-body CT and esophagogastroduodenoscopy performed every six months or the first three years, then annually up to the seventh-year post-treatment.” Was there any loss to follow-up, and if so, how many people were lost to follow-up?
- Results: The P value calculated by the chi-square test for CN stage in Table 2 should actually be 0.033. That is to say, there are differences among the three groups in terms of lymph node metastasis, and the baselines among the three groups are not consistent.
- The article did not explain why hematological toxicity was excluded from the postoperative acute toxicity.
- There was no description of whether there were differences in the incidence of acute toxicity among the three groups.
- Would it be more appropriate to refer to this study, which is based on a retrospective analysis of two-center databases with such a small sample size, as a multi-center retrospective cohort study? Because real-world studies usually require a much larger sample size than RCT studies to have relatively reliable results.
- Figure 1 appears to be somewhat unclear and may benefit from being enhanced for better visibility.
Author Response
please see the detailed responses in the attached file

Reviewer 2 Report
Comments and Suggestions for Authors
Based on the ESOPEC trial, the FLOT regimen appears to be a more effective treatment than the CROSS protocol for patients with non-metastatic esophageal or esophagogastric junction adenocarcinoma. However, this study challenges that conclusion and provides additional real-world data on the subject.
Revised Questions and Comments:
-
Why does your cohort not include patients with esophageal adenocarcinoma (EAC) classified as Siewert type III? Please clarify this point.
-
The 3-year disease-free survival (DFS) and overall survival (OS) are key outcomes of this study. Please revise Table 2 to include both DFS and OS data.
-
Please include a figure presenting 3-year OS across the three cohorts.
-
The figures are unclear. Kindly replace them with high-resolution images.
-
For outcomes related to univariate analysis (e.g., 3-year DFS and OS), please summarize the results in one or two tables to help readers interpret the findings more easily.
-
The 3-year DFS and OS outcomes from your cohorts appear better than those reported in the recent ESOPEC trial. Aside from differences in toxicity profiles, what other factors might explain the superior outcomes in your cohorts? Please elaborate.
Reviewer 3 Report
Comments and Suggestions for Authors
-
In the manuscript, the authors present a study regarding 3 types of different therapies performed in esophageal adenocarcinoma.( standard chemoradiotherapy, intensified chemotherapy followed by RTCHT, and perioperative chemotherapy using the FLOT regimen). It is a multicenter study in which 142 patients were included. In my opinion, it is an interesting study that can be published. In order to improve the quality of the manuscript, some changes have to be done. My observations are :
- please present the postoperative and preoperative morbidity in a more detailed way. Please include some data regarding the preoperative morbidity based on therapy that was used.
- please include some data regarding the treatment modalities that was used in cases of postoperative complications (especially in cases of anastomotic leakage)
Round 2
Reviewer 1 Report
Comments and Suggestions for Authors
The article has been modified accordingly. However, I still have some doubts about the title. If a study has been named in advance and is divided into multiple parts, it should be a registered study. However, the article does not mention the registration information of the study. I hope the author can add this registration information.
Author Response
Dear reviewer,
we plan to perform a second analysis on organ preservation in squamous cell carcinoma, with a retrospective multi-insitutional study covering the same years in our centers, that's the reason for the title. However, our study is not registered on Clinical trials.gov and therefore, if reviewer agree, we can eliminate the terminology "part I" from the title, mention the study only with "A4"
Reviewer 2 Report
Comments and Suggestions for Authors
Thanks for the authors' revision. Everything is fine except that please refine the Table 3 with the same quality of Table 1 and 2 (remove the inner lines of table, make every line equal size..etc).
Author Response
thank you for your comments, we formatted table 3 according other tables style.
Reviewer 3 Report
Comments and Suggestions for Authors
In the manuscript, the authors present a study regarding the efficiency of different therapies performed in esophageal adenocarcinoma.The manuscript has been reviewed before and the authors changed the manuscript according to the precious reviewer indications. Their comments are quite pertinent. That is why, I think that this manuscript can be published in this form.
Author Response
we thank reviewer 3